# Investigation on Purification of Saturated LiNO_3_ Solution Using Titanium Phosphate Ion Exchanger: Kinetics Study

**DOI:** 10.3390/ijms232113416

**Published:** 2022-11-02

**Authors:** Marina Maslova, Vladimir Ivanenko, Polina Evstropova, Natalia Mudruk, Lidia Gerasimova

**Affiliations:** Tananaev Institute of Chemistry—Subdivision of the Federal Research Centre “Kola Science Centre of the Russian Academy of Sciences” (ICT KSC RAS), 184209 Apatity, Russia

**Keywords:** lithium nitrate, titanium phosphate, transition metal ions, sorption, kinetics

## Abstract

Lithium compounds are of high interest to many industries. The presence of undesirable impurities in Li precursors leads to uncontrolled change in the functional properties of final compounds. Therefore, the development of reliable methods for lithium salt purification is considered a key factor for their application in various industries. This work focuses on the application of a titanium phosphate ion exchanger (Li-TiOP) toward Cu^2+^, Co^2+^, Mn^2+^, Ni^2+^, and Cr^3+^ ions in the purification of a saturated LiNO_3_ solution. The sorption kinetics of the selected ions, considering external and internal mass transfer, as well as chemical interaction, were deeply studied. The kinetic study showed that the values of intraparticle diffusion rate and effective diffusion coefficients for the studied ions decreased in the following order: Cr(III) ˃ Cu(II) Mn(II) ˃ Co(II) ˃ Ni(II). For all the selected ions, chemical interaction was described with a pseudo-second-order reaction model. The sorption kinetics were controlled by the size of the solvated metal ion, its effective charge, the electronic structure of the adsorbed ion, and the interaction with the functional groups of the sorbent. Due to fast kinetics, the high degree of removal of trace quantities of the impurities this material gives it consideration as a promising sorbent for the deep purification of lithium salts.

## 1. Introduction

Lithium, due to its unique properties, such as electrochemical activity, high energy density, and specific heat capacity, has become a key element of modern electric-powered vehicles and portable electronic devices [1,2]. During the last decade, the global lithium market has extended rapidly, and the worldwide demand across many cutting-edge industries has increased essentially [3]. One of the main requirements for high-quality lithium-containing functional materials is their purity [4]. Highly purified lithium compounds with purities of >99.9% are used for special ceramics and glass, thermal nuclear systems, acoustic and optical technology, power laser systems, and biomedicine [5,6,7,8,9]. The most undesired impurities for such materials are transition metals. The presence of trace elements can dramatically affect the electrical and optical properties, as well as others. In optical materials, the concentration of transition metals should not exceed 10^−4^ wt.%. In nuclear and semiconductor applications, an impurity content of no more than 10^−10^% is required. For lithium-ion battery (LIB) production, lithium carbonate purity of 99.5% is needed. The occurrence of impurities of transition metals in lithium salt can change the composition and structure of target crystalline cathode or anode materials and lead to uncontrolled change in the functional properties of final compounds. The purification of lithium salt is also very important, taking into account recovery lithium from spent LIBs for the production of new ones.

In recent years, several techniques have been used for transition metal removal from saturated lithium solutions. The simple method for the purification of a metal-polluted solution is the precipitation of Li as LiOH. The influence of the ionic strength and Li concentration in a solution on the solubility of metal hydroxides is the main disadvantage of this method [4]. The solvent extraction method suffers from the operational complexity of technology and the cost of treatment [10,11]. For these purposes, ammonium pyrrolidine dithiocarbamate (APDC) or ethylenediaminetetraacetic acid (EDTA), with higher affinities for the complexation of impure ions, are used as chelating agents. Metal-chelate complexes have lower solubility in the aqueous phase than in the organic phase, so impurities are concentrated in the organic phase, which allows achieving a purification level of 99.99% through four steps of extraction.

The sorption method is a more attractive and easy-to-manage process that does not require additional chemicals. Milutin and coauthors used chelating commercial sorbents, such as Purolite S930, Amberlite IRC 748, and AXIONIT 3S, and obtained results demonstrating that lithium salts (Li_2_CO_3_ and LiCl) of high purity could be obtained [12]. These sorbents manifested selectivity toward certain transition metals and their combination yielded a purification level of 99.90%.

Sorbents based on titanium phosphates (TiPs) are well-known to possess high sorption abilities toward transition metal ions. The sorption capacities of TiPs of different compositions in aqueous solutions were 0.48–1.9, 0.46–0.56, 0.08–0.12, 0.33, and 0.22 mmol·g^−1^ for Cu^2+^, Co^2+^, Ni^2+^, Mn^2+^, and Cr^3+^ ions, respectively [13,14,15,16,17]. In our previous investigation, a synthesized TiO(OH)H_2_PO_4_ ion exchanger showed high efficiencies in wastewater purification: its sorption capacities were (mmol·g^−1^) 1.12–1.60, 1.40, 1.50, and 1.45 for Co^2+^ Ni^2+^, Cu^2+^, and Mn^2+^ ions, respectively [18,19]. In addition, for the first time the sorption of these impurities on Li-substituted TiO(OH)H_2_PO_4_ (Li-TiOP) in a saturated lithium solution was studied. It was found that this material effectively removed trace quantities of metal ions from multicomponent solutions, resulting in a LiNO_3_ purity level of more than 99.999 wt.%. The sorption equilibrium was thoroughly investigated. The selectivity of Li-TiOP toward the studied ions was found to be as follows: Cr^3+^ ˃ Cu^2+^ ˃ Mn^2+^ ˃ Co^2+^ ˃ Ni^2+^ [20].

When using the sorption process in practice, the kinetic characteristics of the sorbent should be taken into account to determine the efficiency of the sorption, which depends on the contact time between the solute and sorbent. Identifying the rate-controlling step is a key factor for selecting the most favorable operating conditions for the practical application of the sorbent. A literature survey showed that studying the kinetic properties of titanium phosphates has not been given due attention. Authors have usually focused on the investigation of sorption steps using chemical adsorption reaction models [21,22,23]. Meanwhile, the diffusion stage can have a marked effect on the overall rate of the sorption process. Thoroughly studying sorption kinetics may give substantial information for the identification of the sorption process.

Therefore, this work is the logical continuation of an investigation on the sorption of transition metals from saturated lithium solutions on titanium phosphate. It is focused on studying the kinetic features of the sorption process, including diffusion and chemical adsorption. To understand the mechanism of sorption kinetics, the values of the ionic potentials of the studied ions and activation energy are calculated. It is shown that the calculation of the ionic potential of the adsorbed ions allows the prediction of their kinetic behavior and sorption efficiency.

## 2. Results and Discussion

### 2.1. Characterization of the Sorbent

The products obtained were X-ray amorphous. The elemental analysis data for as-synthesized titanium phosphate gave a composition of 22.82% Ti and 15.31% P, which corresponded to a molar ratio of Ti:P = 1:1. The calcination at 750 °C resulted in the formation of titanium phosphate with the composition of Ti_2_O(PO_4_)_2_ (Figure 1a).

For Li-substituted titanium phosphate, the final composition could be represented as follows: Ti—21.2%, P—13.7%, and Li—6.13%. The Li^+^ uptake was calculated to be 8.82 meq·g^−1^, which is in very good agreement with that of the IEC calculated from the TiO(OH)Li_2_PO_4_·2H_2_O formula (8.85 meq.g^−1^). The Li-TiOP was also amorphous in nature. This material was thermally unstable, and calcination at 750 °C resulted in the formation of a mixture of phases with compositions of LiTiO(PO_4_), Li_3_PO_4_, and TiO_2_ (Figure 1b).

The measured textural properties of the initial and Li-substituted titanium phosphates show that the materials were obtained as mesoporous sorbents with specific surface areas of 142.4 and 32.87 m^2^·g^−1^ for TiOP and Li-TiOP, respectively. At the same time, the average pore diameter of the synthesized products was in the range of (9.6–8.7 nm, Appendix A).

### 2.2. Sorption Kinetics of Transition Metal Ions on Li-TiOP Sorbent

The gradient of chemical potential is the motivating force for sorption from a solution to a sorbent. When the chemical potentials in a solution and a sorbent are leveled, chemical equilibrium occurs. External mass transfer (transport of a solute to a sorbent surface) or migration of a solute through the porous space of the sorbent (internal diffusion) can be rate-controlling steps. To clarify the rate-controlling stage, kinetic sorption curves were built (Figure 2).

According to results obtained, the adsorption equilibrium was reached in 3–20 min for Cu^2+^ and Cr^3+^ ions and in 60 min for Mn^2+^, Co^2+^, and Ni^2+^ ions. The maximum metal uptake reached 0.2, 0.199, 0.182, 0.164, and 0.158 mmol·g^−1^ for Cu^2+^, Cr^3+^, Mn^2+^, Co^2+^, and Ni^2+^, respectively. Trublet et al. [19] found that the sorption process on the titanium phosphate composition of TiO(OH)H_2_PO_4_·H_2_O was relatively fast, and equilibrium for the studied ions was attained in 10–20 min. In a concentrated lithium solution, the transition metal uptake after 20 min of sorption was estimated to be 95–99% for Cr^3+^ and Cu^2+^ ions and 45–60% for Ni^2+^, Co^2+^, and Mn^2+^ ions. The adsorption depended on interactions between the metal species and the occurrence of active surface sites of Li-TiOP. At a low initial concentration of metal ions, the uptake process included not only the adsorption step, but also the transfer of the metal ions from the electrolyte solution to the sorbent surface.

### 2.3. Diffusion Sorption Kinetics of Transition Metal Ions

To identify the external diffusion step, Boyd’s film diffusion model was applied. The plots of log(1 − *F*) versus *t* fitted to the experimental data are shown in Figure 3a. It is obvious that the presented curves were not straight lines, and curvature of the lines appeared for all the studied temperatures. The experimental data did not satisfy Boyd’s film diffusion model and gave a linear relationship with *R*^2^ ~ 0.68–0.84.

At the same time, it was found that, in the first 5–10 min, the presented curves were straight lines for all the studied ions (Figure 3b). The lines did not intercept the origin, which could be caused by impact of the internal diffusion process. Intraparticle diffusion could be concluded to also be engaged in sorption at the initial stage.

To characterize the sorption kinetics considering the internal diffusion mechanism, Boyd’s model of gel diffusion was applied. According to this model, the plot of *Bt* (calculated by Equation (8)) vs. *t* should produce a straight line that intercepts the origin. The rate constant of intraparticle diffusion *B* can be determined by the slope of the line, and the effective diffusion coefficient *D_i_* can be calculated according to Equation (7). *D_i_* is the kinetic coefficient that characterizes the diffusion transport of the adsorbed ions in the pores of the Li-TiOP sorbent.

For the sorption of Mn^2+^, Co^2+^, and Ni^2+^ ions, the graphs were straight lines that passed through the origin (Figure 4). The experimental data correlated well with the model of gel diffusion kinetics over the entire time, affirming that intraparticle diffusion was the rate-controlling stage.

According to the results obtained, the external diffusion of Cu^2+^ and Cr^3+^ ions occurred in the first 10 min, and the intraparticle diffusion stage was believed to last for the rest of the time. The lines did not intercept the origin (Figure 4), confirming that diffusion occurred from mingled mechanisms. The linear sections of the graphs may be assumed to correspond to intraparticle diffusion, which was preceded by diffusion in the film, and both these types of mass transfer proceeded simultaneously in the initial period.

The values of the rate constants *B* and diffusion coefficients *D_i_* are shown in Table 1. As the lines for copper and chromium ions did not intercept the origin, high errors in estimating *D_i_* were, therefore, observed.

The calculated values of *D_i_* show that diffusion resistance in the pore volume occurred. It could be caused both by steric and electrostatic hindrances for hydrated ions that were adsorbed and their affinities to the sorbent surface. The diffusion coefficients (*D_i_*) for the studied ions decreased in the series of Mn(II) ˃ Co(II) ˃ Ni(II). This trend was due to the values of the solvated metal ions. A similar trend was also observed in the sorption of Mn(II), Co(II), and Ni(II) from a multicomponent solution of lithium nitrate.

It should be assumed that a decrease in the ionic radius of above the ions leads to an increase in their ionic potentials (z/r). The value of the latter determines the size of the solvation shell, which forms in the electric field of the metal ion and is caused by the ion–ion or ion–dipole interactions between metal ions and solvent molecules or background electrolyte ions. An increase in the temperature of a solution leads to increase in the value of the diffusion coefficient due to the partial dehydration of the ion shell and, thus, a decrease in the effective radius. This, in turn, facilitates the sorption of ions on sorbent surfaces.

Based on the effective diffusion coefficients, the values of the activation energy of the sorption process were calculated with the following equation [24]:*D_i_* = *D_o_·exp*(−*E_aKm_*/*RT*),(1)
where *E_a_* is the activation energy, kJ∙mol^−1^; *D* is the constant, m^2^∙s^−1^; *R* is the gas constant, J·mol^−1^; and *T* is the temperature, K.

This equation may be expressed in linear form as follows:log*D_i_* = log*D_o_* − *E_aKm_*/*RT*,(2)

The plots of log*D_i_* vs. 1/*T* were built (Figure 5), and the values of the activation energy were calculated.

The calculated values of the activation energy are presented in Table 1. The values of *E_a_* were in the range of 9–13 kJ·mol^−1^, which confirmed the diffusion of the solute in the sorbent pores [25].

For the sorption of Mn(II), Co(II), and Ni(II) ions, the changes in the values of the activation energy were in good agreement with the changes in the values of the ionic potential (Figure 6).

The calculated *D_i_* values for Cu^2+^ and Cr^3+^ were found to be higher than those for Mn^2+^, Co^2+^, and Ni^2+^. We expected that the diffusion coefficient would be lowest for Cr^3+^ due to the fact that it had the largest charge and the smallest radius among the studied metal ions. Therefore, the ionic potential of the chromium ions should considerably exceed those of the other metal ions, and the hydrated Cr shell should be the largest with the smallest D_i_. However, the diffusion coefficients for the studied ions decrease in the following series: D_Cr_ ˃ D_Cu_ ˃ D_Mn_ ˃ D_Co_ ˃ D_Ni_. The high values of D_i_ for the chromium ions could be attributed to the hydrolysis of Cr^3+^ ions and the formation of an octahedral Cr(OH_2_)_5_(OH)^2+^ species. This was in good agreement with the change in the concentration of hydrogen ions in the chromium-containing solution. A decrease in the effective charges of the resulting cationic forms of chromium ions led to a decrease in the sizes of its hydrated shells, an increase in the mobility of the chromium ions, and finally, an increase in the values of the diffusion coefficients.

Cu^2+^ ions were hydrolyzed under the chosen experimental conditions to a lesser extent. The dominant form of ions was presented in the solution as aquatic copper ions, or Cu^2+^_aq_.

The changes in the values of the effective diffusion coefficients and activation energies for the studied ions were concluded to correlate well with the values of ionic potential (z/r_cr_) of these ions. The smaller the ionic radius of the ion, the greater its potential and, consequently, the larger the solvation shell. A metal ion with a larger solvate shell had lower mobility and more difficulty in ion mass transfer in its pores. An increase in temperature contributed to the partial dehydration of the solvate shells of the metal ions, and an increase in the diffusion coefficients was observed.

### 2.4. Adsorption Reaction Models

In order to reveal the contribution of chemisorption to the overall rate of the sorption process, several adsorption reaction models were applied.

Whether external mass transfer had a notable effect on the sorption process could be adequately described with a pseudo-first-order reaction model. A pseudo-second-order model implies that solute–sorbent interactions and intermolecular interactions of the adsorbed ions occur. The Elovich model takes into account both the sorption and desorption processes and their influences on the kinetics of solute uptake. The parameters related to these models and the *R*^2^ coefficients are present in Table 2.

Following the pseudo-first-order model, the plots of log*(q_e_* − *q_t_)* against time (Figure 7) did not produce a straight line, and linearity deviation could be seen at all the studied temperatures. It can be seen that the values of the sorption capacity *q_e_* calculated by this model largely diverged from the experimental points *q*_exp._, so the experimental data did not satisfy this model.

For Mn(II), Co(II), Ni(II) ions, the values of the rate constant *k*_2_ decreased in the following series: Mn(II) ˃ Co(II) ˃ Ni(II). The values of *k*_2_ increased slightly with the increase in temperature in the same order. The kinetic behavior of these ions was in good accordance with their ionic radius. The smaller the crystal radius, the larger the ionic potential and size of the solvation shell.

It can be seen that, for Cu^2+^ ions, the value of k_2_ was markedly higher than those of other ions. This phenomenon could be related to the difference in the nearest environment of the adsorbed ions. Mn(II), Co(II), and Ni(II) ions formed aqua complexes with octahedral configurations of the inner sphere complexes. According to the Jahn–Teller effect Cu(II) ions decrease in symmetry to keep their stability, and the formation of more stable tetrahedral configurations may be expected. This type of complex is less saturated with ligands and exhibits higher kinetic characteristics during sorption.

For Cr(III) ions, the value of k_2_ was two times lower than that for Cu(II) ions but one order of magnitude higher than those of Mn(II), Co(II), and Ni(II) ions. Chromium(III) ions exhibited a greater tendency toward hydrolysis, with the formation of Cr(OH_2_)_5_(OH)^2+^ species. The decrease in the effective charge of the Cr(III) hydroxy complex led to a reduction of its solvate shell, resulting in an increase in the sorption rate compared to Mn(II), Co(II), and Ni(II) ions. On the other hand, the sorption of Cr(III) required a more complicated reorganization of its internal coordination sphere with the elimination of hydroxyl ligands. This led to slowing of the sorption of Cr(III) compared to Cu(II).

The values of the activation energy of the sorption process could be calculated using the Arrhenius equation [26]:*k* = *A*·*exp*(−*E_a_*/*RT*),(3)
where *E_a_* is the activation energy, *R* is the gas constant, *T* is the absolute temperature, and *A* corresponds to the Arrhenius pre-exponential factor. A plot of log*k*_2_ vs. 1/*T* enabled the determination of *E_a_*/*R* from the slope of the straight line (Figure 9). The data obtained are given in Table 2.

The values of activation energy, adsorption rate constants, and ionic potentials were found to change in following order: Mn(II) > Co(II) > Ni(II). The same decreasing order of sorption rate constants k_2_ were also found to be maintained in a study of the sorption kinetics of Mn(II), Co(II), Ni(II) Cu(II), and Cr(II) ions in a multicomponent solution.

The interpretation of the experimental data for Mn(II), Co(II), Ni(II), and Cu(II) given by the Elovich model (Figure 10, Table 3) shows that plots of *q_t_* versus log*t* gave linear relationships with *R*^2^ > 0.99 in the chosen temperature range.

The linear dependence of *qt* on log*t* indicates that the nature of the sorption did not change as the sorption process proceeded and the sorbent was filled with solute. In addition, hydrated lithium ions (Li^+^_aq_) were replaced by hydrated Mn(II), Co(II), Ni(II), and Cu(II) ions without any change in the energy of the interaction.

For Cr(III) ions, the kinetic curve had two slopes (before and after 10 min of sorption). This indicated a change in the mechanism of sorption and may be associated with a change in the nature of the interaction between the chromium ions and the sorbent phase. The sorption process can involve several simultaneously occurring processes: the partial dehydration of chromium ions, the adsorption of hydrated metal ions, and polarization interactions with a functional group of the sorbent. In this case, the bonds between the functional group and the sorbate ions are stronger than those for the electrostatic interaction. This type of interaction can lead to a decrease in the positive charge on the sorbent surface, which is determined by the content of hydrated lithium ions weakly bound to the sorbent. This creates more favorable conditions for the further sorption of Cr(III) ions. As the sorbent is filled with sorbate, the desorption constant *β* increases, as well as the sorption rate *α* (Table 3).

## 3. Materials and Methods

### 3.1. Materials

Solid oxotitanium sulphate with the formula of TiOSO_4_·H_2_O was used as a titanium precursor, and 10% H_3_PO_4_ solution was used as a precipitant for titanium phosphate synthesis. Industrial-grade Li_2_CO_3_ and analytical-grade HNO_3_, LiOH·H_2_O, Co(NO_3_)_2_·6H_2_O, Mn(NO_3_)_2_·6H_2_O, Cr(NO_3_)_3_·9H_2_O, Cu(NO_3_)_2_·9H_2_O, and Ni(NO_3_)_2_·6H_2_O_2_ were purchased from Neva-Reaktiv (Russia). Stock solutions were diluted with distilled water to prepare required concentrations of the chemicals.

### 3.2. Synthesis of Titanium Phosphate Sorbent (TiOP)

An amount of 100 g of titanium salt was mixed with phosphoric acid solution so the molar ratio of TiO_2_:P_2_O_5_ was equal to 1:0.5. The suspension was heated to 60 °C and kept for 5 h under constant stirring. The precipitate obtained was separated by filtration and washed with H_2_O until washing pH = 3.0–4.0 was reached.

For sorption experiments, the titanium phosphate obtained was mixed with a 0.2 M solution of LiOH to produce the Li-substituted form of titanium phosphate (Li-TiOP). The route of the synthesis was as follows: 100 mL of 0.5 M LiNO_3_ was mixed with 10 g of TiOP, and then 50 mL of 1 M LiOH was added dropwise to the mixture under constant stirring at 60 °C. The suspension was kept under stirring at ambient conditions for 2 h before being filtrated. The resulting material was rinsed with water and dried at 60 °C for 4 h.

### 3.3. Preparation of Saturated LiNO_3_ Solution

A total of 788.6 mL of HNO_3_ with a concentration of 799 g·L^−1^ was mixed with 500 mL of distilled H_2_O. An amount of 369.4 g of Li_2_CO_3_ was gradually added to the solution, and the pH of the solution was about 4. Then, 1 M LiOH was adding to the solution until the pH value reached 5–5.5. Thereby, 5 M LiNO_3_ solution was obtained.

To produce contaminated LiNO_3_ solutions, corresponding Co(NO_3_)_2_, Mn(NO_3_)_2_, Cr(NO_3_)_3_, Cu(NO_3_)_2_, and Ni(NO_3_)_2_ solutions at required concentrations of impurities were added to 5 M LiNO_3_.

### 3.4. Characterization Techniques

Elemental compositions of the sorbents obtained were determined using a Shimadzu ICPE-9000 spectrometer. The thermogravimetric (TG/DTG) and differential scanning calorimetric (DSC) data of samples were collected using a thermogravimetric analyzer (Netzsch STA 409 PC/PG) under an argon atmosphere. XRD data were obtained with a Shimadzu D6000 diffractometer with monochrome *CuK_α_* radiation (λ = 1.5418 Å). The BET surface properties of the samples were determined using a Tristar 320 surface area analyzer. The pore size distribution was calculated using the BJH method. The concentration of transition metals in the filtrates for every sorption experiment was determined using atomic adsorption with an AAS 300 Perkin-Elmer spectrometer. Sorbent particles of 0.1 mm were prepared for study by sieving.

### 3.5. Sorption Experiments

An experimental study of the purification of saturated LiNO_3_ solutions was carried out using a batch technique at 25 °C and pH = 5.5. The concentration of transition metals in the LiNO_3_ was 0.2 mmol·L^−1^. To determine the sorption capacity of the sorbent towards the studied ions, 40 mL of this solution was mixed with 0.2 g of the sorbent and kept under constant stirring at ambient conditions for 24 h to ensure that equilibrium occurred.

The amount of metal ions adsorbed by Li-TiOP was determined by the difference in the concentrations of metal ions in the solution before and after sorption. The amount of metal ions adsorbed (mmol·g^−1^) was calculated with the following formula:(4)$$q_{e}=\frac{C_{0}-C_{e}}{m}\cdot V$$
where *C_o_* and *C_e_* are the initial and final concentrations of the metal ions in the solution (mmol·L^−1^), respectively; *V* is the volume of the solution (L); and *m* is the adsorbent mass (g).

### 3.6. Modeling of Kinetics

The kinetic studies were performed in duplicate using batch experiments at 293–313 K. The initial transition metal concentration was 0.2 mmol·L^−1^. The sorption time for the studied metals was 1–60 min. The sorbent (1 g) was mixed with 200 mL of 5 M LiNO_3_ solution and kept under vigorous stirring (300 rpm) for 1, 3, 5, 10, 20, 30, 40, 50, and 60 min. The chosen temperature was maintained with a thermostat. To determine the maximum amount of metal uptake, the suspensions were kept under stirring conditions for 24 h to ensure that equilibrium occurred. After the sorption process, the concentrations of the transition metals in the solutions were determined.

To describe the sorption kinetics quantitatively, Boyd’s diffusion models (film and intraparticle diffusion) [27], as well as adsorption reaction models [28,29,30], were applied to simulate the kinetics data.

The equation for film diffusion was expressed by following equation [31]:−*ln*(1 − *F*) = 3(*D_i_ct*)/(*rδm*),(5)
where *F* is the fractional attainment at equilibrium, *D_i_* is the coefficient of solute diffusion through a film of thickness *δ*, *r* is radius of the sorbent particle, *t* is the contact time, and *c* and *m* are the concentrations of the sorbate and sorbent in the solution, respectively.

The fractional attainment of equilibrium *F* was calculated using the following formula:*F* = *q_t_*/*q_e_*(6)
where *q_t_* and *q_e_* (mmol·g^−1^) are the amount of the sorbate at any time *t* and at equilibrium, respectively.

The effective diffusion coefficient was calculated with following equation [32]:(7)$$D_{i}=\frac{r^{2\;}}{\pi^{2}}B$$
where *r* is radius of the sorbent particle (m), and the kinetic coefficient *B* can be calculated as follows [33]:*Bt* = −2*F*·log(1 − *F*)(8)

In order to evaluate internal diffusion, Boyd’s model of gel diffusion was applied. The plot of *Bt* (calculated by Equation (8)) vs. time should give a straight line that passes through the origin. Boyd’s coefficient *B* can be determined from the slope of the line, and the effective diffusion coefficient *D_i_* can be calculated according to Equation (7).

The pseudo-first-order model equation was expressed as follows:(9)$$\log(q_{e}\;-\;q_{t})=\log q_{e}-\frac{k_{1}}{2.303}t$$
where *q_e_* and *q_t_* are the amounts of solute adsorbed at equilibrium and at time *t* (mmol·g^−1^), respectively; and *k*_1_ is the rate constant (min^−1^).

The rate constant of the sorption and the amount of metal cation uptake at equilibrium could be determined from a plot of log(*q*_e_ − *q_t_*) vs. *t*.

The pseudo-second-order equation model was described by the following linearized equation:*t*/*q_t_* = 1/(*k*_2_·*q_e_*^2^) + *t*/*q_e_*(10)
with *k*_2_ being the rate constant (g·mmol^−1^·min^−1^).

The values of *q_e_* and *k*_2_ could be determined from the slope and intercept of the plot of *t/q_t_* against *t*.

The Elovich model describes the kinetics of heterogeneous chemisorption on solid surfaces and was expressed by the following equation:*q_t_* = (1/*β*)ln(*αβ*) + (1/*β*)ln*t*(11)
where *α* and *β* are the initial sorption rate constant (mol·g^−1^·min^−1^) and desorption constant (g·mol^−1^), respectively; and *q*_t_ is the sorption capacity at time *t* (mmol·g^−1^).

The constants *α* and *β* were calculated from the slope and the intercept of the plot of *q_e_* vs. ln*t*, respectively.

## 4. Conclusions

For the first time, the sorption kinetics of Mn(II), Co(II), Ni(II), Cu(II), and Cr(III) ions on Li-substituted titanium phosphate in a saturated LiNO_3_ solution with thorough investigation of gel and intraparticle diffusion, as well as chemisorption, were studied.

Boyd’s external and internal diffusion models were applied to obtain insights into the kinetics of ion exchange processes. It was found that diffusion proceeded according to a mixed mechanism. The diffusion coefficients *D_i_* for the studied ions decreased in the series of Cr(III) ˃ Cu(II) Mn(II) ˃ Co(II) ˃ Ni(II).

The values of the rate constant of intraparticle diffusion and the effective diffusion coefficients for Cr(III) ions were significantly higher than those for the other studied ions. The high values of D_i_ for chromium ions could be attributed to the hydrolysis of Cr^3+^ ions and the formation of a Cr(OH)^2+^ hydroxy species. The decrease in the effective charges of the resulting cationic forms of chromium ions led to a decrease in the sizes of its hydrated shells, an increase in the mobility of chromium ions, and finally, an increase in the diffusion coefficients.

The pseudo-second-order model gave a very good description of the sorption kinetic process for all the selected ions. The kinetic behaviors of Mn(II), Co(II), and Ni(II) ions were in a good accordance with their ionic radius values.

The low values of activation energy in the sorption processes of the abovementioned ions according to the pseudo-second-order reaction model indicated that, for these ions, ion exchange proceeded without the essential rearrangement of the nearest environment of the sorbate.

At the same time, low values of diffusion coefficients may indicate considerable difficulty in ion mass transfer in pores or change in the solvate shell. An increase in temperature can contribute to partial dehydration of the solvate shells of metal ions and increase in the value of diffusion coefficients.

For Cu(II) ions, the value of the rate constant *k*_2_ was much higher than that for other ions, which was caused by the tetrahedral configuration of the copper complex. This type of complex was less saturated with ligands and exhibited higher kinetic characteristics in the sorption process.

For Cr(III) ions, the value of k_2_ was two times lower than that for Cu(II) ions but one order of magnitude higher than those for Mn(II), Co(II), and Ni(II) ions. The decrease in the effective charge of the Cr(III) hydroxy complex (Cr(OH)^2+^) led to the reduction of its solvate shell. This, in turn, resulted in an increase in sorption rate compared to Mn(II), Co(II), and Ni(II) ions. On the other hand, the sorption of Cr(III) required a more complicated reorganization of its internal coordination sphere, with the elimination of hydroxyl ligands, which led to the slowing of sorption for Cr(III) compared to Cu(II).

The obtained experimental results on the sorption of the studied metal ions on Li-TiOP in concentrated solution of lithium nitrate allowed us to conclude that the main factors controlling sorption were the following: size of the solvated metal ion, its effective charge, the electronic structure of the adsorbed ion, and the interaction with the functional groups of the sorbent.

The fast kinetics and ability to efficiently adsorb trace quantities of impurities in saturated Li solutions make titanium phosphate a very promising material for lithium salt purification.

## Figures and Tables

**Figure 1 ijms-23-13416-f001:**
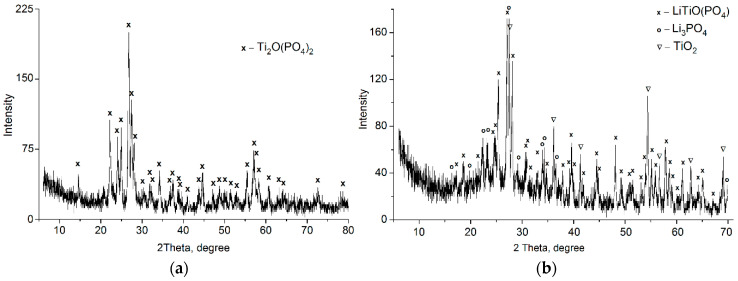
XRD powder patterns of TiOP (**a**) and Li-TiOP (**b**) products calcined at 750 °C.

**Figure 2 ijms-23-13416-f002:**
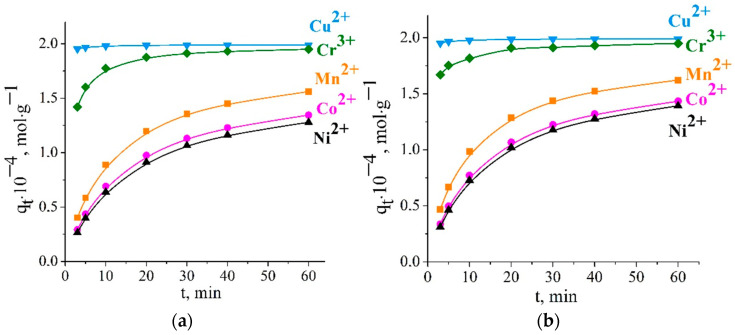
Sorption kinetics of Cu^2+^, Co^2+^, Ni^2+^, Mn^2+^, and Cr^3+^ ions on Li-TiOP at 298 K (**a**) and 313 K (**b**).

**Figure 3 ijms-23-13416-f003:**
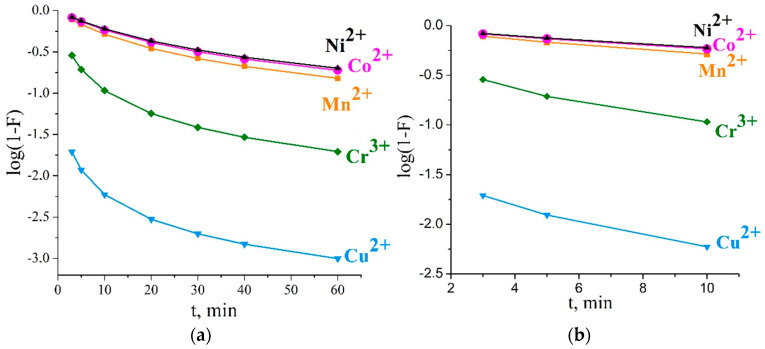
Film diffusion kinetics plots for Cu^2+^, Co^2+^, Ni^2+^, Mn^2+^, and Cr^3+^ sorption on Li-TiOP at 298 K for contact times of 60 min (**a**) and 10 min (**b**).

**Figure 4 ijms-23-13416-f004:**
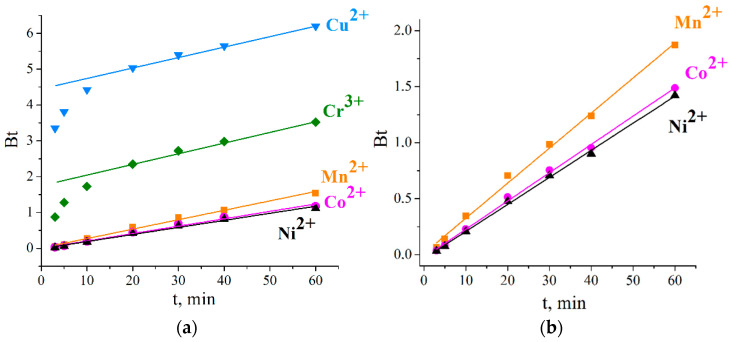
Boyd’s plots for Cu^2+^, Co^2+^, Ni^2+^, Mn^2+^, and Cr^3+^ sorption on Li-TiOP at 298 K (**a**) and 313 K (**b**).

**Figure 5 ijms-23-13416-f005:**
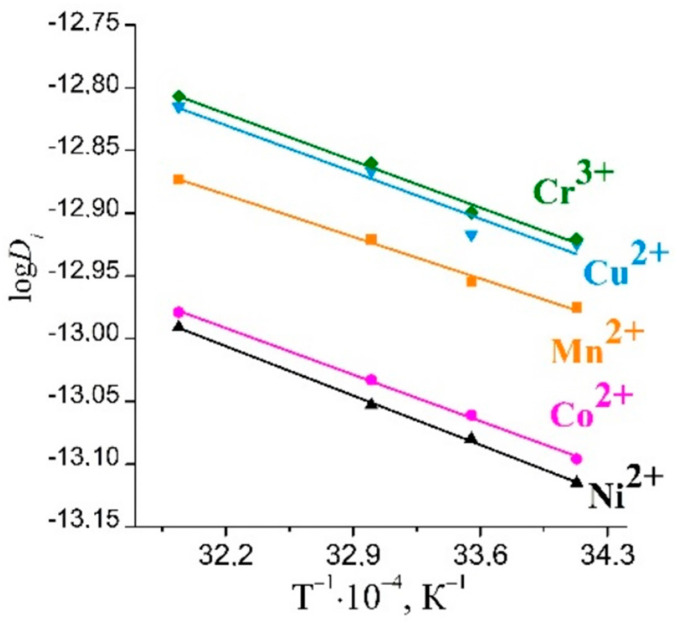
Plot of log*D_i_* versus 1/T.

**Figure 6 ijms-23-13416-f006:**
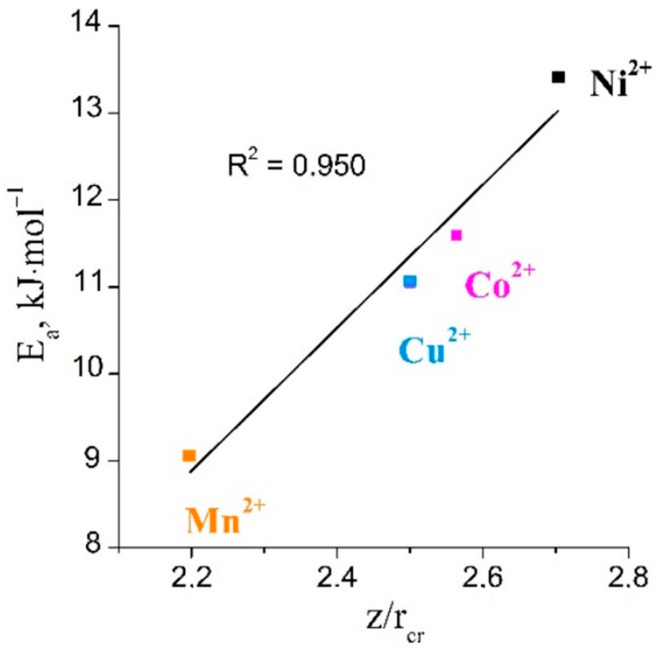
Plot of activation energy versus ionic potential for Mn (II), Co(II), and Ni(II) ions.

**Figure 7 ijms-23-13416-f007:**
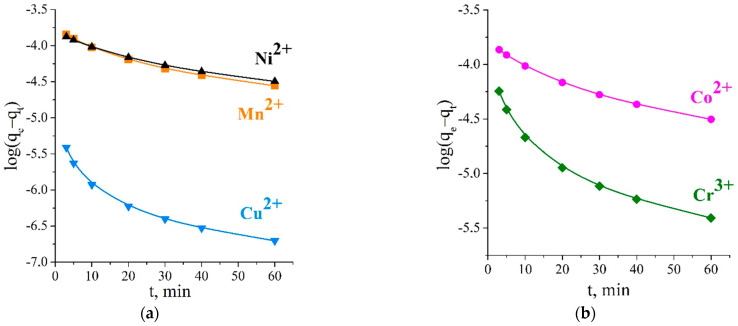
Pseudo-first-order kinetics plot for studied ions’ adsorption on titanium phosphate at 298 K (**a**,**b**) and 303 K (**c**). The kinetic experimental data obtained for the sorption from multicomponent solution also did not correspond to the pseudo-first order model, but the experimental data were fitted well to the pseudo-second-order reaction model. The plots of *t/q_t_* versus *t* were straight lines, and the regression coefficients *R*^2^ were close to 1 for all the studied ions. The obtained calculated values of *q_e_* coincided with the values of *q_exp_* (Figure 8).

**Figure 8 ijms-23-13416-f008:**
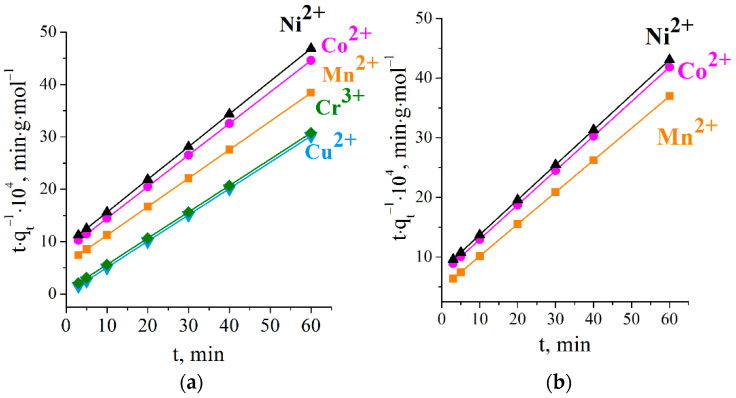
Pseudo-second-order kinetics plot for studied ions’ adsorption on titanium phosphate at 298 K (**a**) and 303 K (**b**).

**Figure 9 ijms-23-13416-f009:**
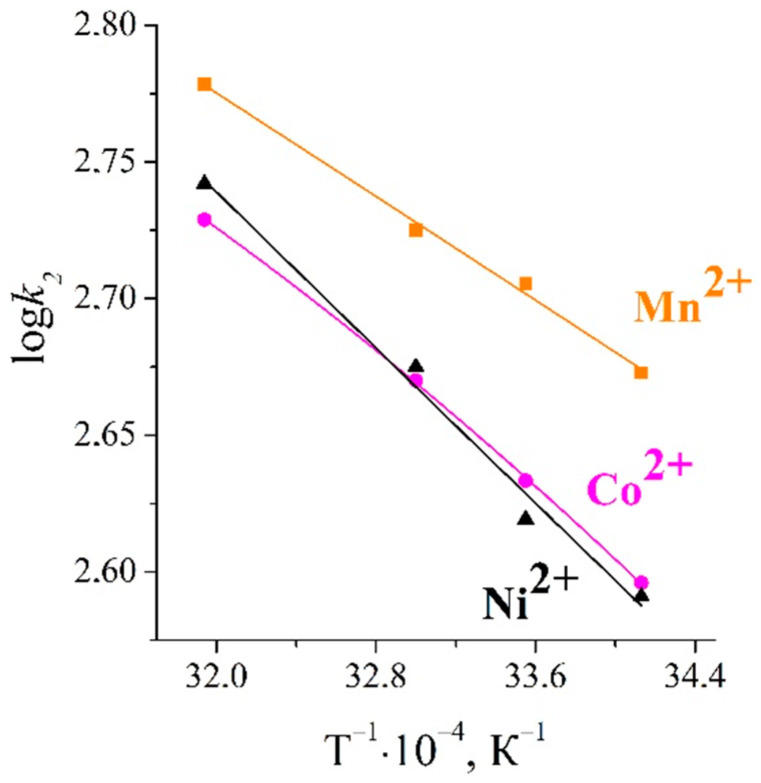
Plot of log*k_2_* vs. 1/*T* for pseudo-second-order kinetics model.

**Figure 10 ijms-23-13416-f010:**
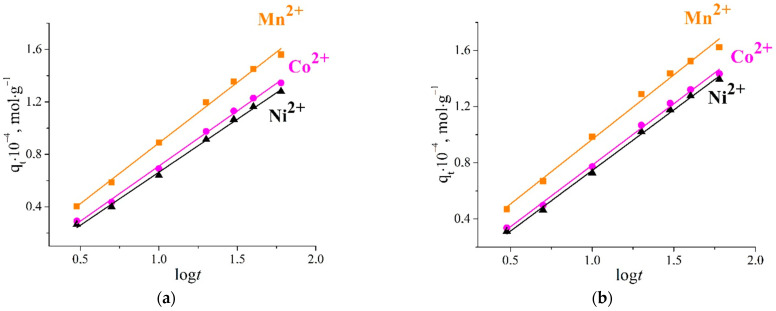
Elovich kinetics plot for adsorption of studied ions on titanium phosphate at 298 K (**a**,**c**) and 303 K (**b**).

**Table 1 ijms-23-13416-t001:** Kinetic parameters of the intraparticle diffusion model for sorption of studied ions on Li-TiOP sorbent at different temperatures.

Me	T, °K	B·10^−2^, min^−1^	R^2^	*D_i_*, m^2^·s^−1^	*E_a_*, kJ·mol^−1^	R^2^
Mn^2+^	293	2.51	0.991	1.06 × 10^−13^	9.12	0.993
	298	2.63	0.993	1.11 × 10^−13^		
	303	2.85	0.998	1.20 × 10^−13^		
	313	3.17	0.996	1.34 × 10^−13^		
Co^2+^	293	1.90	0.996	8.01 × 10^−14^	10.12	0.998
	298	2.06	0.998	8.69 × 10^−14^		
	303	2.20	0.999	9.26 × 10^−14^		
	313	2.49	0.998	1.05 × 10^−13^		
Ni^2+^	293	1.82	0.994	7.67 × 10^−14^	10.83	0.999
	298	1.97	0.990	8.31 × 10^−14^		
	303	2.10	0.995	8.86 × 10^−14^		
	313	2.42	0.999	1.02 × 10^−13^		
Cu^2+^	293	2.74	0.996	1.19 × 10^−13^	11.0	0.995
	298	2.86	0.999	1.21 × 10^−13^		
	303	3.45	0.997	1.36 × 10^−13^		
	313	4.17	0.998	1.53 × 10^−13^		
Cr^2+^	293	2.79	0.997	1.20 × 10^−13^	13.11	0.998
	298	2.88	0.999	1.26 × 10^−13^		
	303	3.47	0.998	1.38 × 10^−13^		
	313	4.19	0.999	1.56 × 10^−13^		

**Table 2 ijms-23-13416-t002:** Kinetic parameters of pseudo-second-order model for sorption of studied ions on Li-TiOP sorbent at different temperatures.

Me	T, K	*q_e_*_(exp)_·10^−3^, mol·g^−1^	*q_e_*, 10^−3^, mol·g^−1^	*k*_2_, g·mol^−1^·min^−1^	R^2^	*E_a_*, kJ·mol^−1^	R^2^
Mn^2+^	293	0.182	0.180	470.76	0.995	9.07	0.996
	298	0.183	0.184	507.62	0.997		
	303	0.184	0.184	530.88	0.990		
	313	0.185	0.187	600.60	0.990		
Co^2+^	293	0.162	0.160	394.45	0.993	11.60	0.998
	298	0.164	0.166	429.91	0.996		
	303	0.167	0.169	467.73	0.990		
	313	0.171	0.173	535.67	0.992		
Ni^2+^	293	0.155	0.153	389.94	0.995	13.41	0.993
	298	0.158	0.160	416.22	0.9956		
	303	0.163	0.168	473.15	0.988		
	313	0.169	0.170	552.16	0.998		
Cu^2+^	293	0.195	0.199	84,030.32	0.995		
	298	0.199	0.199	84,030.32	0.995		
	303	0.199	0.199	84,030.32	0.995		
	313	0.199	0.199	84,030.32	0.995		
Cr^3+^	293	0.199	0.199	4176.10	0.996		
	298	0.199	0.199	4176.10	0.996		
	303	0.207	0.199	4176.10	0.996		
	313	0.213	0.199	4176.10	0.996		

**Table 3 ijms-23-13416-t003:** Kinetic parameters of the Elovich model for sorption of studied ions on Li-TiOP sorbent at different temperatures.

Ion	T, K	*α,* mol·g^−1^·min^−1^	*β*, mol·mg^−1^	R^2^	Comments
Mn^2+^	298	3.52 × 10^−2^	2.61 × 10^−4^	0.996	
	313	4.37 × 10^−2^	2.63 × 10^−4^	0.994	
Co^2+^	298	2.63 × 10^−2^	2.57 × 10^−4^	0.998	
	313	3.06 × 10^−2^	2.62 × 10^−4^	0.998	
Ni^2+^	298	2.61 × 10^−2^	2.87 × 10^−4^	0.998	
	313	2.98 × 10^−2^	2.68 × 10^−4^	0.998	
Cu^2+^	298	2.17 × 10^14^	4.61 × 10^5^	0.973	
Cr^3+^	298	8.15 × 10^−4^	3.29 × 10^−4^	0.989	1st section (before 10 min)
	298	4.35 × 10^14^	2.32 × 10^5^	0.992	2nd section (after 10 min)

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
