# Peer review of "Investigation on Purification of Saturated LiNO_3_ Solution Using Titanium Phosphate Ion Exchanger: Kinetics Study"

_ijms, 2022, doi:10.3390/ijms232113416_

Round 1

Reviewer 1 Report

The main issue that I had with this manuscript was the narrative and the presentation of the concepts.  This journal might have an order of presentation that puts the results ahead of the experimental section, but I think the latter needs to be presented before the former (2. Results and Discussion).  For example F is introduced in line 126, but it is defined after line 356.  The same happens with the models by Boyd and Elovich, explained in equations after lines 366 and 377, respectively, but used in the text. 

Another issue is the scope of the paper.  The authors did work with 5 divalent common transition metals, but they could have expanded to other potential contaminants to Li(NO3) solutions such as divalent Cd, Zn, Ba and Mg. 
Finally I did not find anywhere about the range of contaminating concentrations of elements that were tried.  I assume that concentrations of the 5 divalent ions were varied in order to obtain the second order rate constants, but the experiments were not described and the results were not presented.

There are several other important issues shown below:

1. line 8 should read Li compounds are of high interest to many industries

2. Line 44 should read "precipitation of Li as LiOH"

3. the thermogravimetric analysis shown in lines 90 and 93 does not show the total composition (up to 100%).  Can the authors elaborate on this?

4. Line 126 and thereafter "lg" should be replaced with "log".  As it is it could be confused with natural or Neperian logarithms (ln). The unsaturation fraction (1-F) should have been defined at this point

5. It is clear in Figure 3 that the lines for the kinetics for Cr+3 and Cu+2 do not intersect the y-axis at log(1-F) = 0.  But what happens if the experiment is performed at a temperature close to freezing (~275K)?  If all 5 ions intersect at that temperature, that would indicate the mechanisms of sorption are similar

6. In line 145, would it not be the case that intraparticle diffusion as rate limiting would require no dependence of the rate constants on the ions concentrations, like with enzyme kinetics?

7. In line 155 it should read "a high error in D is therefore observed"

8. In line 162 the authors do not state which value they are referring to: ionic radius, polarizability"

9. Figures 5 and 6 do not have the data points for Cu+2 and Cr+3 but the resulting parameters are shown in Table 1

10.  There is some confusion about Cr+3.  The main species at low pH values is Cr(OH2)6+3, which upon increasing the pH ionizes (in water) to Cr(OH2)5(OH)+2, the rest of the ions, except for Cu+2 should also be octahedral.  Cu+2 is tetrahedral (as explained in the text) according to the Jahn Teller effect.  This should be indicated in lines 200-201.

10. In line 218 the fraction of equilibrium ( lg(qe - qt) ) is used to generate the y-axis in Figure 7, but qe is estimated at equilibrium, which is not really known experimental.  I would think that a better model with -log(qt) and letting the constant log(qe) be and ajustable parameter

11. The Elovich model has an adjustable parameter that is called qe. But this parameter is experimental, calculated according to equation 4.  What is then the difference between the experimental qe which they now call qexp and how is the model qe calculated, given that the model shown in equation 7 shows only qt. This is related to line 221 with the adsorption reaction models

12. In line 241 it should read Jahn-Teller.  The crystal distortion theory has 2 authors.

13. In line 247 it should read Cr(OH2)5(OH)+2

14. in line 256 the number "2" should not be there

15. in line 265 there is no citation to a reference from a previous study.

16. line 269 should read at the chosen temperature range

17. line 276 shows Cr+2 but Figure 10 shows Cr+3.  Please clarify

18. In lines 282-284 one should expect that the hydrate Li interactions with the sorbent should be much weaker than those of all the ions.  So the stronger Cr+3 binding could just be a pure electrostatic interaction over the other 4 cations.  Recall that Cr+3 is a very non-labile metal center since it has a d5 electron configuration that precludes ligand exchange reactions. 

Author Response

Response to reviewer 1

Thank you for taking the time to review the manuscript. We appreciate it very much for your insightful comments and suggestions. All your comments helped us to get our study better and to present the results in the best possible way. We have revised the manuscript according to the comments.

Reviewer comments

The main issue that I had with this manuscript was the narrative and the presentation of the concepts.  This journal might have an order of presentation that puts the results ahead of the experimental section, but I think the latter needs to be presented before the former (2. Results and Discussion).  For example, F is introduced in line 126, but it is defined after line 356.  The same happens with the models by Boyd and Elovich, explained in equations after lines 366 and 377, respectively, but used in the text. 

Authors' response:

The authors agree that “materials and methods” section should be presented before “results and discussion” one. But according to the template of this journal a description of “methods” is followed by the “results” section.

Reviewer comments

Another issue is the scope of the paper.  The authors did work with 5 divalent common transition metals, but they could have expanded to other potential contaminants to Li(NO3) solutions such as divalent Cd, Zn, Ba and Mg. 
Finally I did not find anywhere about the range of contaminating concentrations of elements that were tried.  I assume that concentrations of the 5 divalent ions were varied in order to obtain the second order rate constants, but the experiments were not described and the results were not presented.

Authors' response:

The main attention of research is focused on the purification of solutions from alkali and alkaline metals. Although the most harmful impurities for special glass and ceramics are transition metals. Transition metals contained in the lithium precursor change composition and structure of the ceramic and glass products during heat treatment. In functioning of optical materials for laser-cooling crystal, the transition-metal impurities play a dramatic role reducing the cooling efficiency. The concentration of transition metals in optical materials should not exceed 5·10–5, 2·10–5, 5·10–5, 2·10–5, 5·10–5 wt. % for Cu, Co, Mn, Ni, Cr, respectively.

In our previous investigation we have shown that Li-substituted TiO(OH)H2PO4 (Li-TiOP) effectively removes trace quantities of abovementioned metal ions from saturated lithium solutions that results in LiNO3 purity level higher the required standard. The sorption equilibrium was thoroughly investigated (Ivanenko, V.I.; Maslova, M.V.; Evstropova, P.E.; Gerasimova, L.G. Investigation on purification of saturated LiNO3 solution using titanium phosphate ion-exchanger: Equilibrium study. Trans. Nonferrous Met. Soc. China 2022, accepted 05/07/2022, In Press.). Therefore, this work is a logical continuation of investigation on sorption of transition metals from saturated lithium solutions on titanium phosphate. It is focused on studying the kinetic features of sorption process. We are very grateful to you for the valuable comments and the aim of our future investigation will be the study of sorption behavior of other potential impurities in concentrated lithium solution.

The initial transition metals concentration was 0.2 mmol·L−1 (line 342)

Reviewer comments # 3

  1. the thermogravimetric analysis shown in lines 90 and 93 does not show the total composition (up to 100%).  Can the authors elaborate on this?

Authors' response:

In line 90 and 93 the elemental analysis data but not thermogravimetric analysis has been presented.

Reviewer comments # 5

  1. It is clear in Figure 3 that the lines for the kinetics for Cr+3 and Cu+2 do not intersect the y-axis at log(1-F) = 0. But what happens if the experiment is performed at a temperature close to freezing (~275K)? If all 5 ions intersect at that temperature, that would indicate the mechanisms of sorption are similar.

Authors' response:

Thank you very much for the interesting question. We have not performed these experiments, but we will follow your recommendation in our future study.

Reviewer comments # 6

  1. In line 145, would it not be the case that intraparticle diffusion as rate limiting would require no dependence of the rate constants on the ions concentrations, like with enzyme kinetics?

Authors' response:

In real concentrated lithium solution the concentrations of transition metal ions are quite low that causes certain difficulties in the purification technique. In this study the concentrations of transition metals in the LiNO3 was 0.2 mmol·L–1 and in real solution the concentration of undesired metal ions is about 0.01-0.02 mmol·L–1. In saturated Li solution the electrostatic interaction become very week owing to strong ionic screening effect and values of rate constant at these metal concentrations will be close.

Reviewer comments #10

  1. In line 218 the fraction of equilibrium ( lg(qe - qt) ) is used to generate the y-axis in Figure 7, but qe is estimated at equilibrium, which is not really known experimental.  I would think that a better model with -log(qt) and letting the constant log(qe) be and ajustable parameter.

Authors' response:

qe is calculated value and qexp is experimental data. The latter related to the amount of metal ions adsorbed at equilibrium (section 3.5 and 2.2.,Fig.2)

Reviewer comments #10

There is some confusion about Cr+3.  The main species at low pH values is Cr(OH2)6+3, which upon increasing the pH ionizes (in water) to Cr(OH2)5(OH)+2, the rest of the ions, except for Cu+2 should also be octahedral. Cu+2 is tetrahedral (as explained in the text) according to the Jahn Teller effect.  This should be indicated in lines 200-201

Authors' response:

Thank you very much for your valuable comment, all corrections have been fixed.

Reviewer comments #11

11.The Elovich model has an adjustable parameter that is called qe. But this parameter is experimental, calculated according to equation 4. What is then the difference between the experimental qe which they now call qexp and how is the model qe calculated, given that the model shown in equation 7 shows only qt.This is related to line 221 with the adsorption reaction models

Authors' response:

qt parameter but not qe is used in Elovich model (Eq.11).

We are very grateful to the reviewer for the comments and thorough reading of this manuscript.  All misprints have been fixed (red color in text). The details are listed as follows:

  1. line 8 should read Li compounds are of high interest to many industries

The sentence has been rewritten.

  1. Line 44 should read "precipitation of Li as LiOH"

The sentence has been rewritten.

  1. Line 126 and thereafter "lg" should be replaced with "log". 

The misprint has been corrected

  1. In line 155 it should read "a high error in D is therefore observed"

The sentence has been rewritten.

  1. In line 162 the authors do not state which value they are referring to: ionic radius, polarizability"

The sentence has been corrected.

  1. Figures 5 and 6 do not have the data points for Cu+2and Cr+3but the resulting parameters are shown in Table 1

Thank you for your valuable comments. Fig.5 and Fig.6 have been corrected.

  1. In line 241 it should read Jahn-Teller.  The crystal distortion theory has 2 authors.

The misprint has been corrected

  1. In line 247 it should read Cr(OH2)5(OH)+2

This misprint has been fixed

  1. in line 256 the number "2" should not be there.

The misprint has been corrected

  1. line 269 should read at the chosen temperature range

The sentence has been rewritten.

  1. line 276 shows Cr+2but Figure 10 shows Cr+3

The misprint has been fixed

  1. In lines 282-284 one should expect that the hydrate Li interactions with the sorbent should be much weaker than those of all the ions.  So the stronger Cr+3binding could just be a pure electrostatic interaction over the other 4 cations.  Recall that Cr+3is a very non-labile metal center since it has a d5 electron configuration that precludes ligand exchange reactions. 

Authors' response:

The effective potential of less hydrated ions increases with an increase in the ionic radius and a corresponding decrease in size of the hydrated shell. Thus, the interaction of adsorbed metal cations with functional groups of the sorbent increases, also the polarization effect increases, enhancing the strength of the interaction. Finally, the sorption of metal cations must lead to a decrease in the surface charge due to substitution of least polarizable lithium ions by studied metal ions. The change in the entropy and enthalpy of the sorption process is suggested to correlate with the change in the ionic potential of the sorbent.

To confirm this assumption thermodynamic parameters such as enthalpy (ΔHo), entropy (ΔSo), and the Gibbs free energy (ΔGo) have been calculated.

Table Thermodynamic parameters of the sorption process on titanium phosphate.

Me ion

ΔSo, J/mol×K

ΔHo, kJ/mol

R2

ΔGoav, kJ/mol

Mn2+

169.52

9.72

0.997

–40.80

Co2+

191.26

15.98

0.999

–41.02

Ni2+

202.16

20.63

0.998

–39.61

Cu2+

207.07

14.45

0.999

–47.26

Cr3+

359.27

58.58

0.997

–48.48

The characteristics of the sorption process calculated from the experimental data counts in favour of its endothermic nature, and the change in the free energy indicates that a specific interaction between the sorbate and sorbent occurs. The values of ΔHo and ΔSo of the sorption process increase in the order as follows: Mn2+< Co2+<Ni2+<Cu2+<Cr3+. The increase in entropy is so great that it determines the change in the free energy of the process. The change in the thermodynamic characteristics (ΔSo and ΔHo) of sorption line up with the change in the ionic potential of the adsorbed metal ions.

Dear Reviewer, we are very grateful to you for the valuable comments and thorough reading of this manuscript. All remarks were very helpful for us and we have corrected our text according to your recommendation.

Reviewer 2 Report

1. “Lithium due to its unique properties such as electrochemical activity, high redox potential value and specific heat capacity becomes a key element of modern electric-powered vehicle and portable electronic devices.” This claim is not right. The reason for using Li is because of its light weight and high energy density. In addition, Li has low redox potential. Please check this.

2. “The measured textural properties of the initial and Li-substituted titanium phosphate shows that the materials obtained as a mesoporous sorbent with specific surface area of 101 142.4 and 32.87 m2·g–1 for TiOP and Li-TiOP, respectively” Can the authors specify using which characterization tool, this SSA value and the textural property information is obtained. If it is possible, can the authors provide the distribution of pore size with a figure in the supporting information?

3. Can the authors describe “qt 10-4” in the main text? What is the physical meaning of it? How is it related to kinetics?

4. The authors didn’t mention anything reflected in figure 2b, the scientific insight from it. With a higher temperature, the equilibrium should be reached faster. However, it seems to be similar at the two temperatures.

5. “the Boyd’s film-diffusion model has been applied” Can you provide the model with the equation in the main text? What is the “F”? The authors should provide a definition of it. Without this information, it is hard for readers to understand.

6. what is the Elovich model?

Author Response

Response to reviewer 2

 Thank you for taking the time to review the manuscript. We appreciate it very much for your insightful comments and suggestions. All your comments helped us to get our study better and to present the results in the best possible way. We have revised the manuscript according to the comments.

Reviewer comments #1

“Lithium due to its unique properties such as electrochemical activity, high redox potential value and specific heat capacity becomes a key element of modern electric-powered vehicle and portable electronic devices.” This claim is not right. The reason for using Li is because of its light weight and high energy density. In addition, Li has low redox potential. Please check this.

Authors' response:

We are very grateful to you for the valuable comment. This sentence has been rewritten

Reviewer comments #2

“The measured textural properties of the initial and Li-substituted titanium phosphate shows that the materials obtained as a mesoporous sorbent with specific surface area of 101 142.4 and 32.87 m2·g–1 for TiOP and Li-TiOP, respectively” Can the authors specify using which characterization tool, this SSA value and the textural property information is obtained. If it is possible, can the authors provide the distribution of pore size with a figure in the supporting information?

Authors' response:

The BET surface properties of the samples were determined using a surface area analyzer Tristar 320. The pore size distribution was calculated using the BJH method. The pore size distribution plot is presented in SI.

Reviewer comments #3

Can the authors describe “qt 10-4” in the main text? What is the physical meaning of it? How is it related to kinetics?

Authors' response:

According to the template of this journal description of “methods” section is presented ahead the “results” section that cause some misunderstanding.  In section 3 “Materials and Methods” the kinetics models have been described. qt is the amounts of solute adsorbed at time t.

Reviewer comments #4

The authors didn’t mention anything reflected in figure 2b, the scientific insight from it. With a higher temperature, the equilibrium should be reached faster. However, it seems to be similar at the two temperatures.

Authors' response:

Figures have been rearranged for clarity. In concentrated lithium solution the sorption capacity of sorbent increases slightly with increase in temperature. It may indicate the dehydration of the sorbate and a decrease in the radius of the sorbed ion.

Reviewer comments #5, 6.

“the Boyd’s film-diffusion model has been applied” Can you provide the model with the equation in the main text? What is the “F”? The authors should provide a definition of it. Without this information, it is hard for readers to understand.

what is the Elovich model?

Authors' response:

The description of all kinetics models was given in Section 3.

Dear Reviewer, we are very grateful to you for the valuable comments and thorough reading of this manuscript. All remarks were very helpful for us in correcting our manuscript.
